# Depressive Disorder Recognition Based on Frontal EEG Signals and Deep Learning

**DOI:** 10.3390/s23208639

**Published:** 2023-10-23

**Authors:** Yanting Xu, Hongyang Zhong, Shangyan Ying, Wei Liu, Guibin Chen, Xiaodong Luo, Gang Li

**Affiliations:** 1College of Engineering, Zhejiang Normal University, Jinhua 321004, China; xuyanting@zjnu.edu.cn (Y.X.); kalyan2001@zjnu.edu.cn (S.Y.); 2College of Computer Science and Technology, Zhejiang Normal University, Jinhua 321004, China; zhonghongyang@zjnu.edu.cn (H.Z.); liuwei@zjnu.edu.cn (W.L.); cgbcvip@zjnu.edu.cn (G.C.); 3The Second Hospital of Jinhua, Jinhua 321016, China; 4College of Mathematical Medicine, Zhejiang Normal University, Jinhua 321004, China

**Keywords:** depressive disorder (DD), electroencephalogram (EEG), beta rhythm, convolutional neural network (CNN), long short-term memory (LSTM), deep learning

## Abstract

Depressive disorder (DD) has become one of the most common mental diseases, seriously endangering both the affected person’s psychological and physical health. Nowadays, a DD diagnosis mainly relies on the experience of clinical psychiatrists and subjective scales, lacking objective, accurate, practical, and automatic diagnosis technologies. Recently, electroencephalogram (EEG) signals have been widely applied for DD diagnosis, but mainly with high-density EEG, which can severely limit the efficiency of the EEG data acquisition and reduce the practicability of diagnostic techniques. The current study attempts to achieve accurate and practical DD diagnoses based on combining frontal six-channel electroencephalogram (EEG) signals and deep learning models. To this end, 10 min clinical resting-state EEG signals were collected from 41 DD patients and 34 healthy controls (HCs). Two deep learning models, multi-resolution convolutional neural network (MRCNN) combined with long short-term memory (LSTM) (named MRCNN-LSTM) and MRCNN combined with residual squeeze and excitation (RSE) (named MRCNN-RSE), were proposed for DD recognition. The results of this study showed that the higher EEG frequency band obtained the better classification performance for DD diagnosis. The MRCNN-RSE model achieved the highest classification accuracy of 98.48 ± 0.22% with 8–30 Hz EEG signals. These findings indicated that the proposed analytical framework can provide an accurate and practical strategy for DD diagnosis, as well as essential theoretical and technical support for the treatment and efficacy evaluation of DD.

## 1. Introduction

Depressive disorder (DD) is characterized by depressed mood, lack of interest, and loss of pleasure, accompanied by corresponding changes in thinking and behavior [1,2,3]. It is estimated that DD affects more than 300 million people worldwide and covers a wide range of people [4]. According to the World Health Organization, DD is the largest single factor of global disability [5]. As a common mental disease, DD substantially jeopardizes people’s regular life, family, and daily work [6,7,8]. A report in The Lancet claimed that the ratio of DD increased from 9.7% in 2019 to 19.8% in 2020 [9]. The onset of DD is related to a variety of factors, including genetics, environment, individual experience, physiological factors, and gender differences. Previous studies have reported a consistently higher incidence of DD for females than for males [10,11]. Patients with DD lack self-cognition. It is difficult for them to realize that they are suffering from complex psychiatric disorders like DD and may engage in risky behaviors as a result. However, if DD is diagnosed in time and correctly, patients can receive treatment and obtain good curative effects [12]. To sum up, the accurate and practical diagnosis of DD is crucial for patients.

The majority of current diagnostic techniques of DD rely on subjective scale assessments and the clinical expertise of professional psychiatrists according to the diagnostic criteria of DD, such as the Diagnostic and Statistical Manual of Mental Disorders (DSM-5). Psychiatrists can understand the symptoms and manifestations of patients with DD [13,14], including psychological, emotional, and behavioral changes, through these subjective assessment methods. It is well known that different psychiatrists may have different diagnostic results for the same patient with DD, and the diagnostic accuracy varies greatly and is highly subjective [15,16]. To make the diagnosis process of DD more objective and accurate, intelligent diagnostic technology has become a research hotspot [17,18].

Neuroimaging techniques have been widely used to explore the alterations in brain functions in recent years, such as electroencephalogram (EEG) [19], magnetoencephalogram (MEG) [20], functional magnetic resonance imaging (fMRI) [21], etc. These brain imaging techniques have also been applied to understanding the neuromechanisms and realizing the automated intelligent diagnosis of mental disorders [22,23,24]. Due to the advantages of being non-invasive, economical, and easy to operate, the EEG technique has high research and application values in brain-science-related studies [25,26]. It has been reported that EEG signals have apparent changes in different frequency bands and regions in patients with DD [27,28,29]. Based on our brain functional mechanism of DD [19], it has been found that the important neuro-electrophysiological characteristics of DD are mainly distributed in the frontal region of the brain. Meanwhile, the response effects of antidepressant drugs are also related to the dynamic change in EEG power in the frontal region [30,31]. To sum up, this study attempted to achieve high DD recognition accuracy using frontal six-channel EEG signals in combination with deep learning algorithms to improve the practicability of DD diagnosis [32].

Deep learning algorithms are evolving rapidly and are widely used in various fields. A deep learning framework can automatically extract features from EEG signals and eliminate the constraints of artificial features. Deep learning can effectively improve the generalization ability of classifiers, which have been widely used in DD diagnosis research [33,34]. Among them, a convolutional neural network (CNN) can learn and extract feature representations that are robust to input data [35], which is the core of the current best architecture for processing data. Acharya et al. [36] used a CNN to detect DD with EEG features and obtained high accuracy. The long short-term memory network (LSTM) is another commonly used deep learning algorithm which has shown excellent performance with many time series data. Combining these two types of networks for EEG signals, Betul et al. developed a deep hybrid model based on CNN-LSTM architecture to classify the EEG signals of left and right hemispheres and obtained accuracies of 99.12% and 97.66%, respectively [37]. In brief, deep learning algorithms have a good application prospect in DD recognition research.

This study attempted to achieve a high-accuracy and practical model for DD diagnosis with frontal six-channel EEG data. Based on prior research, two deep learning models, multi-resolution CNN (MRCNN) combined with LSTM (named MRCNN-LSTM) and MRCNN combined with residual squeeze and excitation (RSE) (named MRCNN-RSE), were proposed for comparison. Both of these models involved MRCNN to extract the time–frequency domain aspects of EEG features, but two different strategies were used for further extraction and the processing of the extracted features. In addition, the classification performance of each rhythm is also discussed in this study to verify the significant change in beta rhythm in DD patients.

## 2. Materials and Methods

### 2.1. Subjects

All DD patients, excluding those with depressive episodes due to bipolar disorder, were collected from the designated hospital for psychosis, and the HCs were selected from the local community after a professional screening. We randomly recruited 41 patients with DD (10 males and 31 females, respectively). This is consistent with the existing reports that the prevalence of depression is higher in females than in males. Meanwhile, in order to retain this original imbalance the in control group, we randomly selected 34 HCs (11 males and 23 females, respectively). All participants completed the Hamilton Depression scale (HAMD) before EEG data collection. All the HCs had a HAMD score of less than 7, while DD patients had a HAMD score of more than 17. All included subjects were right-handed and prohibited from drinking alcohol and taking psychotropic drugs for 8 h before EEG recording. The age of patients with DD ranged from 19 to 61 years old, with an average age of 45.22 ± 11.80 years old. The age of the HCs ranged from 21 to 57 years, with a mean age of 40.18 ± 11.67 years. There was no significant difference in age between the DD group and the HC group, and there was a significant difference in HAMD-17 scores, as shown in Figure 1 for basic information. The experiment was approved by the Ethics Committee of Zhejiang Normal University, and all participants signed a written informed consent form before the experiment.

### 2.2. Data Acquisition and Preprocessing

In the study, we collected 10 min of resting-state 16-channel EEG data from each subject. The EEG acquisition device used in this study was an EEG TS215605 from Nicolet Company. The EEG channels’ names and positions are shown in Figure 2. The subjects were asked to sit in a chair in a comfortable sitting position with their eyes closed and their attention focused on their breathing. Data acquisition was arranged in the professional EEG lab. The whole EEG collection was implemented in a quiet environment. As is well known, the installation time for EEG recording is highly related to the number of electrodes, especially for non-specialists. An increase in the number of electrodes will naturally lead to high test complexity, high analysis difficulty, and high time and economic costs, which can constrain the practical applications of EEG-related products and systems, particularly to the detriment of large-scale DD early-screening applications in schools and communities. Based on our previous research on DD and the consideration of the accuracy and practicability of the algorithm, this study selected six frontal EEG electrodes (shown in Figure 2) for DD diagnosis.

During the acquisition of EEG signals, it is easy to be disturbed by various factors such as the environment, self-physiologies, and body movements. These noises can negatively affect the signal quality. Preprocessing is a very important step in EEG data analysis which can effectively improve the signal-to-noise ratio of the EEG signal and provide a reliable basis for subsequent analysis and interpretation. The specific steps of EEG preprocessing in this study are as follows:(1)Downsampling

Downsampling refers to a reduction in the high sampling rate to a lower sampling rate for the EEG signal, which mainly aims to reduce the amount of data and improve computational efficiency. In this study, the original EEG signal sampling rate was reduced from 250 Hz to 125 Hz.

(2)Baseline Correction

The main purpose of baseline correction is to eliminate the direct current (DC) offset generated by the recorded signal, which affects the accuracy and comparability of the signals. The baseline correction performed on the EEG signal can remove the DC offset to make the mean of the signal become zero.

(3)Artifact Removal

The purpose of artifact removal is to improve the signal-to-noise ratio and to better reveal the information contained in the EEG signal. The common artifacts in EEG signals include electromyography artifacts, electrocardiograph artifacts, eye movement artifacts, head movement artifacts, etc. In this study, independent component analysis (ICA) was used to remove the artifacts. 

(4)Data segmentation

In this study, 4 s continuous EEG data were selected as the sample without data superposition, resulting in 9354 samples for the DD group and 7443 samples for the HC group.

(5)Filtering

Bandpass filtering is used to remove unwanted frequency components from the signal, preserving the signal in a specific frequency range. By setting the cut-off frequency, the bandpass filter filters out the signals below or above the frequency, and only the signals within the range are retained. In this study, a 4-order Butterworth bandpass filter was applied to the EEG data to extract the specific frequency range, such as theta (4–8 Hz), alpha1 (8–10 Hz), alpha2 (10–13 Hz), beta (13–30 Hz), 4–30 Hz, 8–30 Hz, and 10–30 Hz. 

After the above five preprocessing procedures, the extracted 4 s continuous EEG data were set as the inputs of the proposed deep learning models to explore the DD diagnosis accuracies of the different frequency ranges. Particularly, among these frequency ranges, the 4–30 HZ EEG signals were called the original data because they included all rhythms in this frequency range.

### 2.3. Deep Learning Model Framework

In this current study, two deep learning models, MRCNN-LSTM and MRCNN-RSE, were proposed for DD diagnosis. These two models were improved and derived from the reports of previous studies that CNN-LSTM and CNN are the most commonly used basic model frameworks in the DD detection field. In order to make the results more persuasive, we selected four representative deep learning models according to the recent references and applied them to our EEG data. These four models did not change their structures; they only changed their parameters to fit our data. The specific descriptions of these models are as follows.

#### 2.3.1. MRCNN-LSTM Model Framework

The MRCNN-LSTM model structure is shown in Figure 3, and it mainly includes two parts: MRCNN and LSTM. The first part of MRCNN uses three branches with different convolution kernel sizes in parallel to extract EEG features with the CNN model. Each branch uses a convolution kernel with different sizes to extract features with different scales, and uses the ReLU activation function for nonlinear transformation to enhance feature expressiveness. In the second part of LSTM, the features are spliced and input into an LSTM layer to extract the timing information from the EEG features. The LSTM layer remembers the previous state through a long short-term memory unit with a gating mechanism and updates the state according to the new information input to capture the temporal dependence of the data. The output of the LSTM layer is fed into a fully connected layer and a Dropout layer to improve the generalization ability of the model and suppress overfitting. Finally, binary classification is implemented by a softmax classifier to predict the labels of the inputs.

#### 2.3.2. MRCNN-RSE Model Framework

In accordance with the existing research results, a CNN model with multi-resolution convolution kernels (MRCNN) is proposed in this study. As shown in Figure 4, the MRCNN model contains two branches with two different convolution kernels. Three convolutional layers and two maximum pooling layers are used in each branch. The convolution kernels used in the first branch are set as 4, 3, and 3, respectively. The convolution kernels of the second branch are set as 10, 3, and 3, respectively. In addition, each convolutional layer includes a normalization layer for normalizing the input data, and a Gaussian error linear unit (GELU) is used as the activation function; it is smoother than the traditional ReLU activation function and can better deal with nonlinear features.

To improve the learning performance of the MRCNN model, a calibration module was designed for the features extracted by the convolutional layer to model the interdependence between the features. The residual squeeze-and-excitation (RSE) block was used in the MRCNN model, which is named MRCNN-RSE for short, to adaptively select the most discriminative features. Specifically, the RSE block adaptively selects and readjusts features, helping the model to better utilize contextual information in its local sensory field. In the calibration module, two convolutional layers with a kernel and step size of 1 are used to further extract features, the adaptive pooling layer is used to compress the features, and then two fully connected layers are used to aggregate the information. The first layer uses the ReLU activation function to reduce the dimension, and the second layer uses the sigmoid activation function to increase the dimension. The specific structure for the RSE block is shown in Figure 5.

To extract the interdependence between the extracted features in the MRCNN-RSE model [38], the widely used self-attention mechanism is included in the model, which can adaptively weigh the features of each position for the input data. Specifically, features from each location interact with features from other locations to extract more global context information. In this way, the model can better capture key features for the input data to improve the model performance by assigning higher weights to regions of interest and lower weights to regions of less interest. There are two Add and Normalize layers after the features are weighted by the self-attention mechanism. Finally, the softmax layer is used as the decision function.

#### 2.3.3. Other Model Frameworks

(1)EEGNet

EEGNet is a compact convolutional neural network for EEG analysis. The EEGNet algorithm has better generalization ability and higher performance with limited training data. Liu et al. applied this deep learning framework, EEGNet, to depression diagnosis [39]. The framework consists of four main blocks: convolution, depthwise convolution, separable convolution, and classification. In the convolution block, batch normalization is added. In the depthwise convolutional block and separable convolutional block, batch normalization, activation, average pooling, and Dropout are added. Finally, in the classification block, the two categories, the DD group and the HC group, are identified directly using a fully connected layer.

(2)DeprNet

Ayan Seal et al. proposed a deep learning model based on the convolutional neural network, named DeprNet, for DD detection with EEG signals [40]. The DeprNet model consists of five convolutional layers, five batch normalization layers, five max pooling layers, and three fully connected layers. The softmax activation function is used in the last fully connected layer and the leaky rectified linear unit (LeakyReLU) activation function is used in all other layers, and finally, the classification is conducted using three fully connected layers.

(3)1DCNN-LSTM

Mumtaz et al. proposed a deep learning model combining one-dimensional CNN (1DCNN) with LSTM (named as 1DCNN-LSTM) to detect depression [34]. The model of 1DCNN-LSTM is a cascaded formation of three 1D convolutional layers and two LSTM layers. Max pooling and Dropout are embedded under each convolutional layer, and finally, classification is performed using a fully connected layer.

(4)2DCNN-LSTM

Zhang et al. proposed a 2DCNN-LSTM model to analyze the 128-channel EEG signals for DD detection [35]. The model consists of four 2D convolutional layers and one LSTM layer, with an activation function (Tanh) added after each 2D convolutional layer. Max pooling is accessed after the second 2D convolutional layer for dimensionality reduction. Dropout is accessed after the LSTM, and finally, the classification is performed using a fully connected layer. Compared with the 1D convolutional layer, the 2D convolutional layer can extract features in the spatiotemporal dimension, making it more capable for feature extraction.

### 2.4. Model Evaluation

The confusion matrix is a matrix used to evaluate the performance of a classification model, where each row represents the predicted class and each column represents the actual class, as shown in Table 1. Each cell in the confusion matrix contains the number of samples for the actual and predicted categories that are likely to be classified correctly (true positive, TP; true negative, TN) or incorrectly (false positive, FP; false negative, FN). More precisely, TP represents the number of samples that are positive and correctly predicted to be positive; FP represents the number of samples that are negative but incorrectly predicted to be positive; FN represents the number of samples that are positive but incorrectly predicted to be negative; TN represents the number of samples that are negative and correctly predicted to be negative. Based on the confusion matrix, a series of evaluation indices can be calculated, namely accuracy, precision, recall, and F1_score. As shown in Formulas (1)–(4), accuracy refers to the proportion of correctly classified samples in the total samples; precision refers to the proportion of correctly predicted positive samples in all predicted positive samples; recall refers to the proportion of correctly predicted positive samples in all positive samples; and F1_score is the harmonic average of the accuracy rate and the recall rate.
(1)$$Accuracy=\frac{TP+TN}{TP+TN+FP+FN}$$
(2)$$Precision=\frac{TP}{TP+FP}$$
(3)$$Recall=\frac{TP}{TP+FN}$$
(4)$$F_{1}=\frac{2TP}{2TP+FP+FN}$$

In addition, for the training parameters of all deep learning models in this study, batch_size was set as 32 and the maximum number of epochs was set as 200. A warmup learning rate strategy was used, with an initial learning rate strategy of 5e-5, reaching 1e-3 after 20 rounds, and then gradually decaying to 5e-4, and the weight_decay was set as 0.001. Additionally, five cross-validations were used for all models to reduce the risk of model overfitting and improve the generalization ability of the model.

## 3. Results

### 3.1. The Results of DD Classification Based on the MRCNN-LSTM Model

The results of DD classification based on the MRCNN-LSTM model are shown in Table 2. The average accuracy rate of five cross-validation results is 95.34 ± 0.41%. Meanwhile, the accuracy of the theta rhythm, alpha1 rhythm, alpha2 rhythm, and beta rhythm are 76.14 ± 0.81%, 76.90 ± 0.41%, 80.28 ± 0.66%, and 92.03 ± 0.37%, respectively. These results indicate that a higher frequency band has a higher accuracy for DD recognition.

### 3.2. The Results of DD Classification Based on the MRCNN-RSE Model

As shown in Table 3 and Figure 6, the classification results based on the MRCNN-RSE model have an accuracy of 98.47 ± 0.38% in DD recognition. Compared with the MRCNN-LSTM model in Table 2, the accuracies of the MRCNN-RSE are significantly improved among all EEG rhythms and original EEG data. In addition, the DD recognition accuracy of the beta rhythm is also higher than those of the other rhythms. As shown in Figure 6, the results based on the MRCNN-RSE model have very smooth classification performances after 100 epochs.

### 3.3. Classification Performances for DD Diagnosis with Different Frequency Bands Based on MRCNN-RSE Model

The results of the classification performances for DD diagnosis with 4–30 Hz, 8–30 Hz, 10–30 Hz, and 13–30 Hz EEG signals are shown in Table 4 and Figure 7. It is shown that 8–30 Hz obtained slightly better classification performances for DD diagnosis compared with 4–30 Hz, with higher Accuracy and a lower standard deviation. In addition, the desired classification performance was also obtained for 10–30 Hz, which was not statistically different from the 4–30 Hz results. However, the 10–30 Hz classification performance was significantly lower. The above results show that we can use higher-frequency bands in the DD automatic diagnostic system, which can effectively improve the efficiency of EEG signal preprocessing.

### 3.4. Classification Performances for DD Diagnosis with Different Models Using 8–30 Hz EEG Signals

It has been shown that 8–30 Hz EEG signals obtained the best classification performances based on the MRCNN-RSE model. In this section, the performances of the EEGNet model, the DeprNet model, the 1DCNN-LSTM model, the 2DCNN-LSTM model, and the MRCNN-LSTM model are compared with that of the MRCNN-RSE model on the same 8–30 HZ EEG dataset. These models had the same training settings. The results of the classification performances for DD diagnosis with these models are shown in Table 5. It is shown that the average Accuracy of the EEGNet model, the DeprNet model, the 1DCNN-LSTM model, the 2DCNN-LSTM model, the MRCNN-LSTM model, and the MRCNN-RSE model on the same dataset is 90.07 ± 0.81%, 75.31 ± 0.45%, 87.37 ± 0.65%, 89.75 ± 0.29%, 95.38 ± 0.21%, and 98.47 ± 0.3%, respectively. The MRCNN-RES model still has the best performance among these deep learning models.

## 4. Discussion

Previous research indicated that individuals with DD have altered functional connectivity in the frontal cortex [19]. Based on the findings of our preceding research, this study proposed to use frontal six-channel EEG signals in conjunction with a deep learning algorithm to diagnose DD, which considerably simplifies data collection efforts and improves the practicability of DD screening. It was discovered that the beta rhythm has a greater accuracy than theta, alpha1, and alpha2 rhythms, which indicated that beta had a significant alteration in patients with DD. Simultaneously, MRCNN-RSE achieved the highest accuracy of 98.48 ± 0.22% with 8–30 Hz EEG signals. A detailed discussion is presented below.

### 4.1. Frontal Six-Channel EEG Signals Combined with Deep Learning Show an Excellent Performance for DD Diagnosis

DD is highly related to abnormal brain functions and has dramatically altered functional connectivity in the frontal area of the brain [41,42,43,44]. The frontal cortex is known to play an important role in emotional cognition and working memory [45]. Bludau et al. reported that patients with DD had a significantly smaller medial frontal pole compared with HCs, which was significantly negatively correlated with the severity and course of DD [43]. Coryell et al. evaluated the volume of the left side of the frontal cortex in 10 DD patients and 10 HCs and concluded that patients with severe depressive disorder were more likely to have an increase in the posterior frontal cortex [46]. There is rising evidence that functional connections in the frontal regions of DD are dramatically altered [19]. Therefore, this study advocated for DD diagnosis using the brain’s six frontal EEG channels in conjunction with a current sophisticated deep learning algorithm to attain high accuracy and practicability.

Recently, deep learning has had better predictive performance compared to traditional machine learning for diagnosing depression [33,47]. Qu et al. [48] performed DD identification on a dataset of 2546 veterans using deep learning and five other traditional machine learning algorithms. The results showed that deep learning is more accurate in identifying DD and its risk factors compared to traditional machine learning by ranking the key factors of veterans and capturing the hidden pattern multilayer network structure in the data to obtain better classification performances. Kour et al. [49] combined feature extraction techniques and a hybrid deep learning model of CNN-LSTM for depression classification and compared it with four traditional machine learning models for efficiency comparison, and showed that the recognition accuracy on the benchmark dataset reached 96.78%, which is better than the state-of-the-art traditional machine learning techniques. Deep learning, with its technological advantages, has a stronger predictive ability in DD diagnosis [50]; therefore, this study advocates for the use of deep learning algorithms for the automatic diagnosis of DD.

A previous study has demonstrated that more EEG channels for deep learning can achieve high accuracy and few EEG channels may significantly reduce the accuracy of DD diagnosis. It has been reported that Zhu et al. collected the resting state of 128 EEG channels of 27 DD patients and 28 HCs, and finally achieved an accuracy of 96.50% for DD and HC classification with their proposed CNN model [51]. Yang et al. proposed a gated temporal-separable attention network for EEG-based DD recognition, and the classification accuracy on the EDRA dataset with 62 EEG channels and MODMA dataset with 128 EEG channels was 98.33% and 97.56%, respectively [52]. Wei Liu et al. used a CNN combined with gate-controlled loop units to extract sequence features and obtained an accuracy of 89.63% on the publicly available dataset with 128 EEG channels [53]. In general, more interaction information can be extracted for higher accuracy using high-density EEG. However, high-density EEG (for example, 128 channels) can severely limit the efficiency of data acquisition and reduce the practicability of the model. Some researchers have attempted to detect DD with several EEG channels. For example, Cai et al. employed a three-channel EEG collection system to gather EEG data from the FP1, FP2, and FPz electrodes; nonetheless, the accuracy was only 78.24% [54]. In this study, a deep learning model of MRCNN-RSE that performs well was proposed to identify DD with the frontal six-channel EEG signals, and the accuracy was 98.47 ± 0.01%, which is close to those results of high-density EEG, indicating that the strategy in this study has apparent advantages in classifying DD. 

### 4.2. Beta Rhythm Is Significantly Alerted in DD

EEG signals can be divided into different rhythms according to their frequency and amplitude [55], such as delta, theta, alpha, beta, and gamma, which is an effective approach for psychiatric diseases in EEG-related studies [56]. Studies have shown that EEG signals change significantly with increasing levels of depression [57]. Henriques et al. [58] collected EEG signals from 15 clinically depressed patients and 13 healthy individuals in resting state with eyes closed and found that the activity in the left frontal lobe of depressed patients was significantly lower than that of healthy individuals. Omel et al. found that the activity of the left frontal lobe was significantly lower than that of healthy individuals by examining EEG data [59]. It was found that the regional differences between the anterior and posterior subdivisions of the brain in DD were reduced, and the difference in activity in the left hemisphere relative to the right hemisphere was significant, while the possession attribute of the EEG in depressed patients significantly reduced the relative strength of alpha and beta rhythmic activity. Meanwhile, Knott et al. collected EEG signals from 70 depressed patients and 23 healthy individuals and found that DD patients had relatively reduced overall left hemisphere activity and generally lower delta, theta, alpha, and beta coherence indices. They achieved a classification accuracy of 91.3% for patients and controls [60]. There are significant differences in EEG signals between DD patients and HCs, and brain activity is affected by depression throughout the cerebral cortex.

The results of this study showed that the accuracy obtained in beta rhythm was significantly higher than those in theta rhythm, alpha1 rhythm, and alpha2 rhythm for DD recognition, which revealed that beta rhythm was significantly changed in patients with DD. It is well known that beta rhythms are commonly associated with brain alertness, attentional states, and emotions [19,61]. It has been found that beta EEG power is increased in DD patients compared to HCs and plays an important role in the pathogenesis of DD [62]. Our previous study also found significant changes in the EEG features of power spectral density, fuzzy entropy, and phase lag index of beta rhythms among DD patients, and concluded that the characteristics of beta rhythms play a crucial role in identifying DD [19]. This study further revealed the significant changes in beta rhythms of DD patients through deep learning algorithms, which provided additional important technical support for the diagnosis, treatment, and efficacy assessment of DD.

### 4.3. Practical High-Frequency EEG Signals for DD Diagnosis: Evidence from Deep Learning

In the current study, we attempted to explore the effect of different EEG frequency bands on the classification performances of DD diagnosis. It was discovered that 8–30 Hz EEG signals obtained superior accuracy in DD diagnosis compared to 4–30 Hz EEG data. High-frequency bands yielded good classification performance in a cognitive impairment diagnostic study [63], with higher accuracy than full-frequency bands. Shalini Mahato et al. reported that the accuracy of DD diagnosis can be improved by using different frequency combinations [64]. The results of previous studies have showed that frequency has a large impact on DD classification performance [19]. Our results further confirm this conclusion and extended that high- and wide-frequency EEG signals are better for DD classification. More meaningfully, the use of high-frequency-band EEG data can effectively improve the signal-to-noise ratio and reduce the preprocessing time of EEG data. The findings of this study are of great significance for the development of automatic DD diagnosis.

### 4.4. Limitations

In this study, we achieved exciting DD recognition performance using only a few frontal EEG signals and accelerated practical application of DD automated diagnostics, but this study still has one shortcoming. This research included a limited number of participants, with 41 individuals for the DD group and 34 individuals for the HC group, which may be unable to accurately assess the generalization performance of the MRCNN-RSE model. In future studies, we will continue to gather samples to promote the practical implementation of the classification approach used in this work.

## 5. Conclusions

In this study, we proposed a technology framework for DD precision recognition based on frontal six-channel EEG data and deep learning models. The MRCNN-RSE model achieved a high accuracy of 98.48 ± 0.22% with 8–30 Hz EEG signals and was significantly more accurate than other deep learning models, which is consistent with our previous study using 16-channel EEG signals, indicating that this framework based on frontal EEG signals combined with the MRCNN-RSE model for DD diagnosis is accurate and practical. Our findings can provide a basic theory and technological support and greatly promote the practicality and accuracy of DD diagnosis and efficacy evaluation.

## Figures and Tables

**Figure 1 sensors-23-08639-f001:**
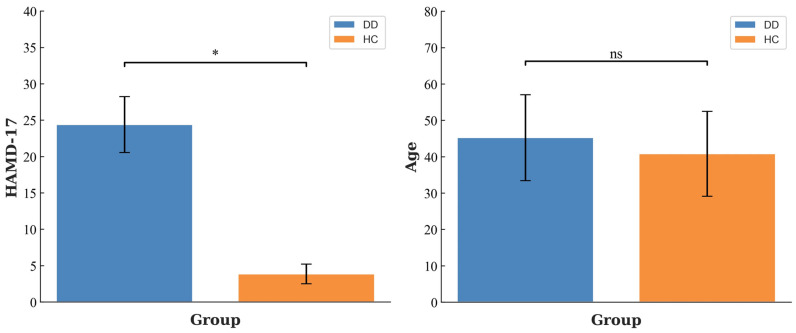
Clinical characteristics of DD and HC. * means *p* < 0.05, and ns means non significant.

**Figure 2 sensors-23-08639-f002:**
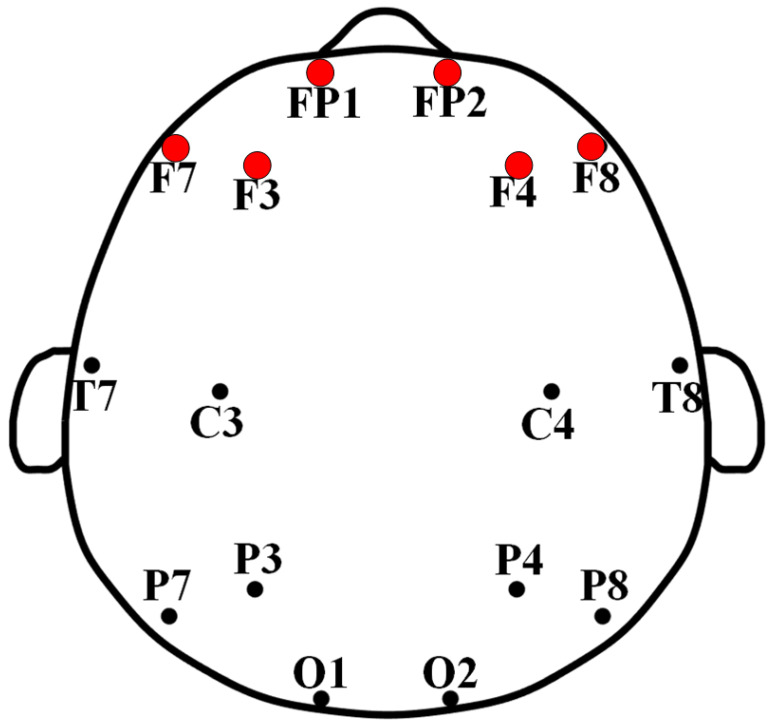
Sixteen EEG channels’ names and locations; red dots indicate the frontal six channels selected for this study.

**Figure 3 sensors-23-08639-f003:**
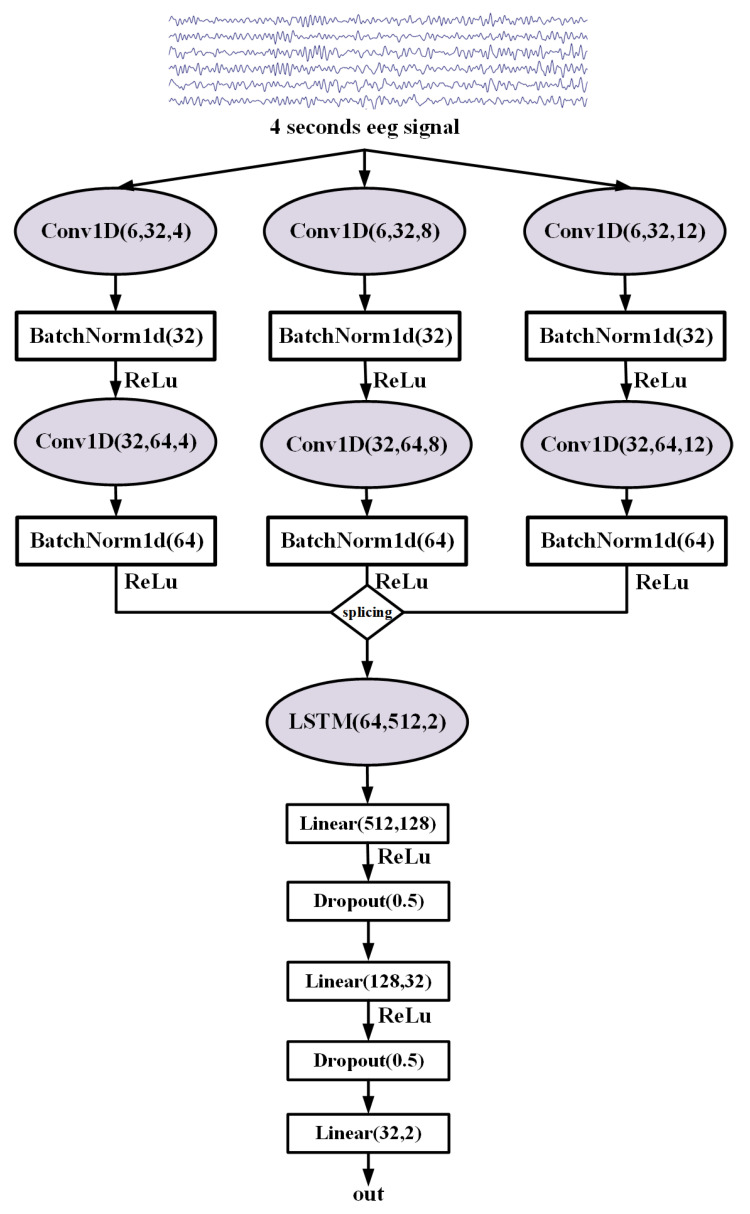
MRCNN-LSTM model architecture.

**Figure 4 sensors-23-08639-f004:**
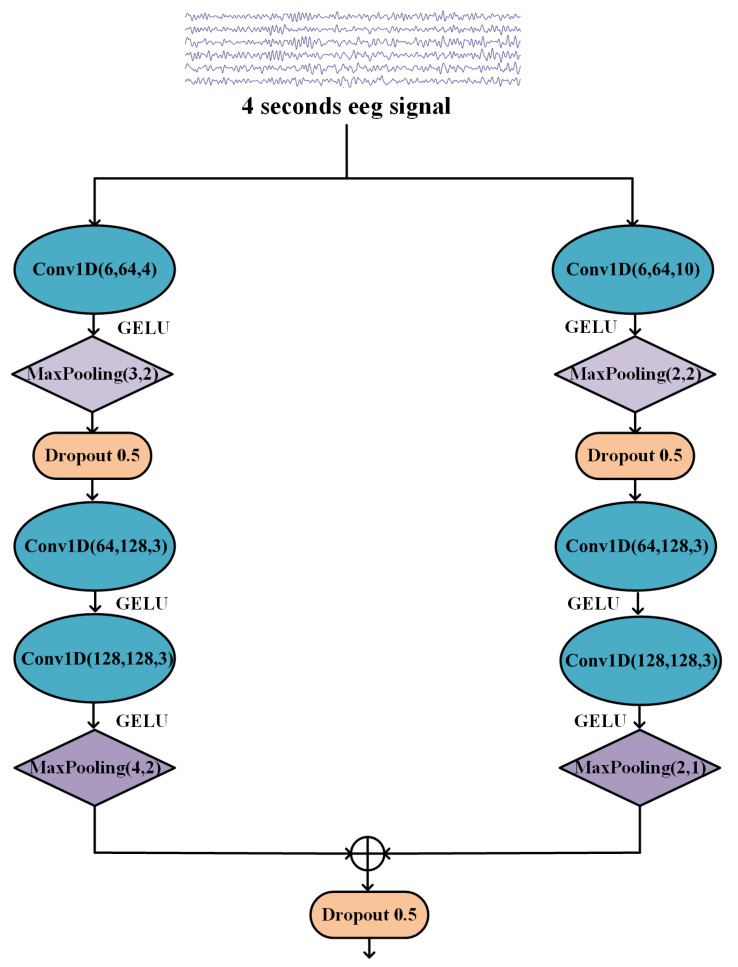
MRCNN architecture.

**Figure 5 sensors-23-08639-f005:**
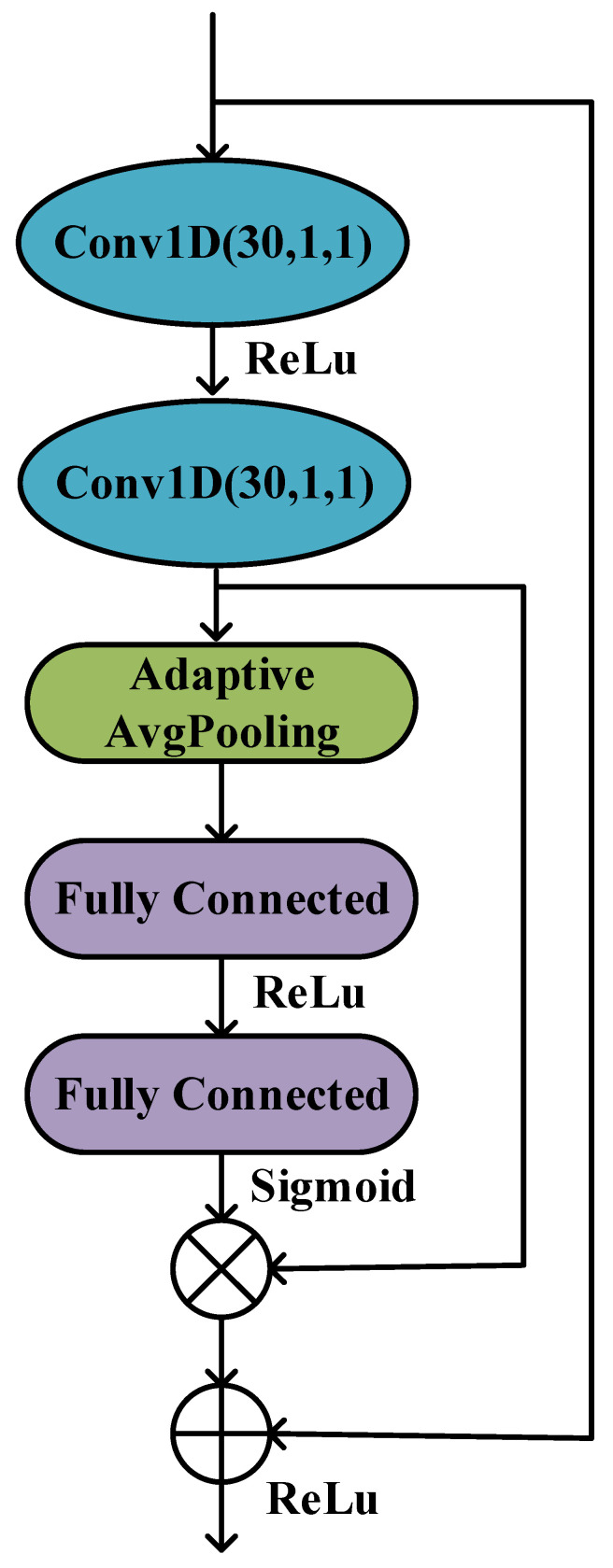
Feature calibration module with RSE block in MRCNN-RSE model.

**Figure 6 sensors-23-08639-f006:**
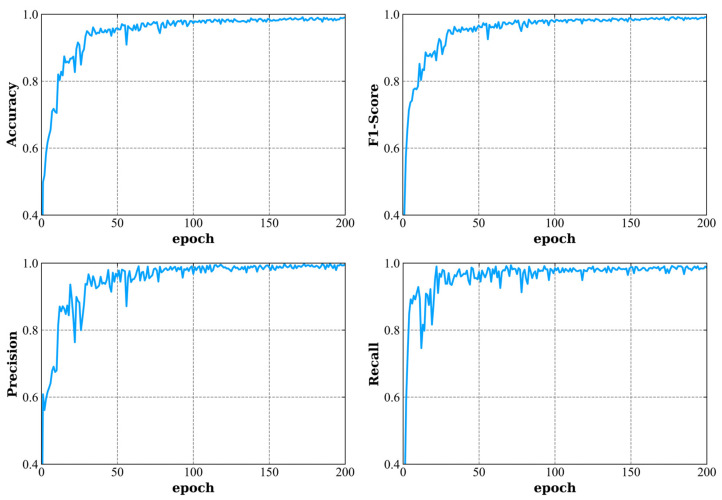
Accuracy, precision, recall, and weighted F1-score using EEG to identify patients with depressive disorder.

**Figure 7 sensors-23-08639-f007:**
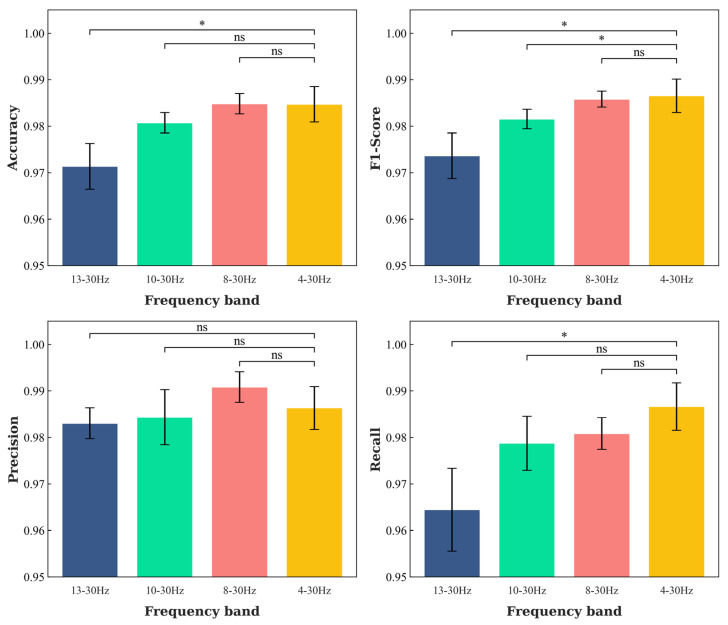
The comparison results of 4–30 Hz, 8–30 Hz, 10–30 Hz, and 13–30 Hz EEG signals for DD diagnosis based on MRCNN-RSE model. * represents *p* < 0.05, and ns represents no statistical difference between the two groups.

**Table 1 sensors-23-08639-t001:** Confusion matrix.

	Actual Class	HC	DD
Predicted Class			
**HC**		*TP*	*FN*
**DD**		*FP*	*TN*

**Table 2 sensors-23-08639-t002:** The results of DD identification among four rhythms and the original data based on the MRCNN-LSTM model.

Data	Accuracy	F1_Score	Precision	Recall
Theta	76.14 ± 0.81%	78.29 ± 1.16%	79.14 ± 1.45%	77.59 ± 3.28%
Alpha1	76.90 ± 0.41%	79.33 ± 0.37%	78.82 ±1.97%	79.98 ± 2.48%
Alpha2	80.28 ± 0.66%	82.56 ± 0.45%	80.98 ± 0.96%	84.21 ± 0.65%
Beta	92.03 ± 0.37%	92.78 ± 0.34%	93.27 ± 1.33%	92.33 ± 1.29%
Original Data	95.34 ± 0.41%	95.79 ± 0.33%	96.03 ± 1.44%	95.57 ± 1.10%

**Table 3 sensors-23-08639-t003:** DD identification results for the four rhythms and original data based on the MRCNN-RSE model.

Data	Accuracy	F1_Score	Precision	Recall
Theta	80.75 ± 0.78%	82.64 ± 0.62%	83.17 ± 1.63%	82.18 ± 1.82%
Alpha1	79.15 ± 1.07%	81.31 ± 1.19%	81.24 ± 0.67%	81.42 ± 2.45%
Alpha2	83.27 ± 0.53%	84.73 ± 0.59%	85.92 ± 1.38%	83.63 ± 2.12%
Beta	97.13 ± 0.49%	97.36 ± 0.49%	98.30 ± 0.33%	96.44 ± 0.89%
Original Data	98.47 ± 0.38%	98.65 ± 0.36%	98.63 ± 0.46%	98.66 ± 0.51%

**Table 4 sensors-23-08639-t004:** DD identification results with different frequency bands based on MRCNN-RSE model.

Frequency Band	Accuracy	F1_Score	Precision	Recall
13–30 Hz	97.13 ± 0.49%	97.36 ± 0.49%	98.30 ± 0.33%	96.44 ± 0.89%
10–30 Hz	98.07 ± 0.22%	98.15 ± 0.21%	98.43 ± 0.59%	97.87 ± 0.58%
8–30 Hz	98.48 ± 0.22%	98.58 ± 0.17%	99.08 ± 0.33%	98.08 ± 0.34%
4–30 Hz	98.47 ± 0.38%	98.65 ± 0.36%	98.63 ± 0.46%	98.66 ± 0.51%

**Table 5 sensors-23-08639-t005:** The results of classification performances for DD diagnosis with different models using 8–30 Hz EEG signals.

Model	Accuracy	F1_Score	Precision	Recall
EEGNet [39]	90.07 ± 0.81%	90.65 ± 0.90%	90.55 ± 1.48%	90.83 ± 2.62%
DeprNet [40]	75.31 ± 0.45%	77.16 ± 1.66%	78.70 ± 2.73%	76.09 ± 5.47%
1DCNN-LSTM [34]	87.37 ± 0.65%	88.54 ± 0.48%	87.59 ± 2.75%	89.72 ± 3.21%
2DCNN-LSTM [35]	89.75 ± 0.29%	90.43 ± 1.53%	90.35 ± 0.80%	90.37 ± 0.41%
MRCNN-LSTM	95.38 ± 0.21%	95.66 ± 0.24%	95.49 ± 0.31%	95.84 ± 0.46%
MRCNN-RSE	98.47 ± 0.38%	98.65 ± 0.36%	98.63 ± 0.46%	98.66 ± 0.51%

## Data Availability

Not applicable.

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
