# Peer review of "Depressive Disorder Recognition Based on Frontal EEG Signals and Deep Learning"

_sensors, 2023, doi:10.3390/s23208639_

Round 1
Reviewer 1 Report
This is an interesting clinical field study to demonstrate the relevance of EEG patterns to the diagnosis of depressive disorder. The authors reveal high levels of accuracy and precision in determination of the diagnosis based on frontal EEG signals and deep learning. One major concern is with the distinction between major depressive disorder and bipolar disorder. The implication of this study design is that authors have targeted major depressive episode and measure state-dependent biomarkers within the active episode, regardless to the nature of the nosological class, which is controversial anyway. While I do share such approach, authors are still advised to more comprehensively justify it under the 2.1. "Subjects" section.
Minor comments: authors may consider other earlier EEG studies in the field, to be incorporated in the Discussion such as e.g. https://pubmed.ncbi.nlm.nih.gov/28976896/
Reviewer 2 Report
Depressive Disorder Recognition Based on Frontal EEG Signals and Deep Learning
In this article, the authors explore the use of EEG signals for the objective and accurate diagnosis of depressive disorder. The paper presents in a clear way methodological approach, however, there are several issues that must be addressed before the manuscript can be considered for publication:
- The gender factor is rather biased to the female participation for the DD participants, but no justification is provided for that. The pool of participants should be created based on the characteristics of the specific disease. For example, the DD is more frequent in females than in males? The authors must include a clear justification for this.
- The EEG equipment that was used should be reported for future reference.
- The authors state that: “cross-validation is used for MRCNN_RSE model”. What about the MRCNN-LSTM?
- The authors must clearly describe the experimental procedure and the obtained results. What is “original data”? It is only mentioned in the results table.
- The authors should include a comparative study, presenting all related studies and their own, and qualitative/quantitative comments.
